# Novel Cuboid-like Crystalline Complexes (CLCCs), Photon Emission, Fluorescent Fibers, and Bright Red Fabrics of Eu^3+^ Complexes Adjusted by Amphiphilic Molecules

**DOI:** 10.3390/polym14050905

**Published:** 2022-02-24

**Authors:** Qinglin Tang, Shasha Liu, Jin Liu, Yao Wang, Yanxin Wang, Shichao Wang, Zhonglin Du, Linjun Huang, Laurence A. Belfiore, Jianguo Tang

**Affiliations:** 1National Center of International Joint Research for Hybrid Materials Technology, Institute of Hybrid Materials, National Base of International Sci. & Tech. Cooperation on Hybrid Materials, College of Materials Science and Engineering, Qingdao University, 308 Ningxia Road, Qingdao 266071, China; tql948889643@163.com (Q.T.); l1274328589@163.com (S.L.); liujin0620@163.com (J.L.); wangyaoqdu@126.com (Y.W.); yanxin_2008@126.com (Y.W.); wangsc@qdu.edu.cn (S.W.); duzhonglin@qdu.edu.cn (Z.D.); huanglinjun@qdu.edu.cn (L.H.); belfiore@engr.colostate.edu (L.A.B.); 2Department of Chemical and Biological Engineering, Colorado State University, Fort Collins, CO 80523, USA

**Keywords:** Eu (III) complex, cuboid-like crystalline, amphiphilic molecules, fluorescent fiber, melt spinning, LDPE

## Abstract

With the growing needs for flexible fluorescence emission materials, emission fibers and related wearable fabrics with bright emission properties have become key factors for wearable applications. In this article, novel cuboid-like crystals of Eu^3+^ complexes were generated. Except for light-energy-harvesting ligands of thenoyltrifluoroacetone (TTA) and 1,10-phenanthroline hydrate (Phen), the crystal structures were adjusted by other functional amphiphilic molecules. Not only does ETPC-SA, adjusted by stearic acid, have a regular cuboid-like crystal with a size of about 2 μm size, but it also generates the best photon emission property, with a fluorescence quantum yield of 98.4% fluorescence quantum yield in this report. Furthermore, we succeeded in producing novel fluorescent fibers by mini-twin-screw extrusion, and it was easy to form bright red fabrics, which are equipped with strong fluorescence intensity, flexibility, and a smooth hand feeling, with the normal fabricating method in our work. It is worth noting that ETPC-HQ fibers, which carry a crystal complex adjusted by hydroquinone, possess the lowest quantum yield but have the longest average fluorescence lifetime of 1259 µs. This result means that a low-density polyethylene (LDPE) matrix could make excited electrons stand in the excited state for a relatively long time when adjusted by hydroquinone, so as to increase the afterglow property of fluorescent fibers.

## 1. Introduction

Fibers and fabrics with bright luminescence seem extremely interesting, and we can even imagine the luminescence signals to be sensed and probed. Most lanthanide elements, such as europium (Eu^3+^) and terbium (Tb^3+^), have unique luminescence performance, a high color purity, and a stable emission wavelength [1,2,3,4,5,6], so they are used in lasers and luminescent materials, as well as biomedical applications [7,8,9,10,11,12]. However, it is a fact that they emit relatively weak luminescence, so lanthanide ions (Ln^3+^) have to be coordinated with organic ligands to form luminescence complexes to obtain better fluorescent properties [13,14,15,16,17]. 

Organic ligands, such as β-diketone, transfer energy to Ln^3+^ to emit characteristic fluorescence [18], which has only ever been used as a laser by Lempicki and Samelson [19]. As for other ligands, aliphatic compounds with carboxylic groups can also be coordinated with Ln^3+^ to form complexes, but they do not usually emit fluorescence. However, Ln^3+^ ions with two or three different kinds of ligands will form different possible complex structures [20,21,22,23].

Polymeric materials have excellent performance and a range of qualities, such as flexibility, thermal stability, and mechanical properties; thus, researchers have attempted to incorporate luminescent complexes into polymer hosts to form composite luminescent materials with both emissions and polymeric flexibility [24]. Based on the structural characteristics of both hydrophobicity from conjugate moiety and hydrophilicity from the polar functional group (i.e., the carboxyl group), designed organic ligands can have energy-harvesting properties and form a complex with Ln^3+^ ions [25,26]. With this characteristic, the conjugated ligand–Ln^3+^ complexes can dope the polymer host to form luminescent fibers, such as Eu^3+^-complex-doped methyl methacrylate (PMMA), for night wear and fire emergency and entertainment clothing applications [27]. In recent years, Guo [28] reported synthesized Ln^3+^-strontium aluminate luminescent fibers using special spinning technology with a matrix of polyester or nylon. For this purpose, the key goal is to develop compatibility between polar metal ions and a nonpolar polymeric host so as to improve the dispersion of the Ln^3+^ complex and the luminescence performance of related hybrid materials [29,30]. 

The photophysical properties, such as photon emission, of organic molecule–Ln^3+^ complexes also depend on the influences of their surrounding environment, formed by themselves and host polymers. Studies designing the surrounding environment have indicated good examples of inorganic matrices. For example, Eu^3+^ inorganic compounds existing as micro-crystals with multiple forms of morphologies, such as spherical, hexagonal plate, diamond, four-angled star, butterfly-shaped, and cuboid morphologies, have been successfully synthesized [31].

In this paper, we focus on the surrounding environmental design of organic ligand–Eu^3+^ complexes in crystals. Similarly to inorganic crystals, organic molecule crystals also have periodic atomic arrangements in unit cells and have low relaxation energy to dissipate, which is helpful to avoid the occurrence of non-radiation de-excitation of excited electrons that have absorbed incident photon energy [32]. Therefore, the crystalline organic ligand–Ln^3+^ structure is a very important aspect to obtain a high photon quantum yield. From this point of view, the crystal shape and size and the related crystal systems are interesting topics for us. In this work, we report a cuboid-like crystal, whose non-spherical shape and high specific surface area ensure the ease of dispersion in the host polymer, with easily sheared down small pieces. This highly efficient dispersion of organic–Eu^3+^ complexes is helpful to emit high-intensity photons to produce shiny fibers and fabrics. Notably, low-density polyethylene (LDPE)—an excellent matrix—provides advantages such as low water absorption, excellent chemical stability, heat fusion, and good machine-shaping properties. Thus, it is a good candidate to be the host material to form high-quality fluorescence fibers and fabrics.

The second important novelty of our work is the amphiphilic organic molecules used to adjust the morphological structure and photophysical data of the cuboid-like crystalline complexes (CLCCs). Stearic acid and hydroquinone are both crucial for the control of morphological structure, and they do not weaken the fluorescence intensity of Eu^3+^ complexes. However, stearic acid and hydroquinone are also excellent auxiliaries of compatibility, which result in a significant ease of making red-emitting fluorescent fabrics with flexibility and a smooth hand feeling. Moreover, the color of the complexes is different when doping with different amphiphilic organic molecules. Therefore, the bright fluorescent fibers and fabrics can be applied to the areas of wearable sensors, probers, indicators, and other wider areas.

## 2. Materials and Methods

### 2.1. Chemicals

Ethanol (AR > 99.7%) was used as solvent to prepare the complexes. EuCl_3_·6H_2_O (99.99%) was purchased from Desheng New Materials CO., Ltd. (Ezhou, Shandong, China) and was used without further purification. 2-thenoyltrifluoroacetone (TTA) (98.0%) and 1,10-phenanthroline (Phen) (99%) were purchased from Macklin Biochemical Co., Ltd. (Shanghai, China) and were used without further purification. Ammonia solution (GR) was diluted to 1ml/L in deionized water. Stearic acid (SA) and hydroquinone (HQ) were used in ethanol solution without further purification. 

### 2.2. Synthesis of Eu^3+^-TTA-Phen Complexes (ETPC)

EuCl_3_·6H_2_O, TTA, and Phen were dissolved in ethanol to prepare a 0.1mol/L solution. They were then mixed at a molar mass ratio of 1:1:1. Meanwhile, ammonia solution of 1 mol/L was added dropwise into the mixture to adjust the pH value from 7 to 8. The mixture was stirred at 35–40 °C for 4.5–5 h. Finally, we obtained complex suspension for further experimentation [33]. The general structure of this complex is displayed in Figure 1(1).

### 2.3. Synthesis of Eu^3+^-TTA-Phen-SA (ETPC-SA)

Stearic acid particle was dissolved in ethanol to prepare a 0.1mol/L solution. Then, it was added to the Eu^3+^-TTA-Phen suspension at a molar mass ratio of 2:1. The operating steps mentioned in Section 2.2 were repeated to obtain the Eu^3+^-TTA-Phen-stearic acid suspension for further experimentation. The general structure of this complex is displayed in Figure 1(2). 

### 2.4. Synthesis of Eu^3+^-TTA-Phen-HQ (ETPC-HQ)

Hydroquinone powder was dissolved in ethanol to prepare a 0.1 mol/L solution. Then, it was added to the Eu^3+^-TTA-Phen suspension at a molar mass ratio of 2:1. The operating steps mentioned in Section 2.2 were repeated to obtain the Eu^3+^-TTA-Phen-HQ suspension for further experimentation. The general structure of this complex is displayed in Figure 1(3).

Figure 1 indicates the structures of the CLCCs, namely, ETPC, ETPC-SA, and ETPC-HQ. TTA and Phen were coordinated at a ratio of 1:1 in ETPC complex, as shown in Figure 1(1). TTA, Phen, and SA were coordinated at a ratio of 1:1:2 in ETPC- SA complex, as shown in Figure 1(2). TTA, Phen, and HQ were coordinated at a ratio of 1:1:2 in ETPC- HQ complex, as shown in Figure 1(3). 

### 2.5. Preparations of Hybrid Fluorescent Fibers Doped with ETPC, ETPC-SA, and ETPC-HQ (CLCC Fibers, CLCCF)

Highly efficient red-emitting hybrid fluorescence fibers were prepared using HAAKE Rheomex PTW16 (Thermo Fisher Scientific, Karlsruhe, Germany). Firstly, CLCC powders were uniformly mixed with host matrix (LDPE) particles in a clean container, resulting in the proportion of CLCC occupying LPDE being about 3 wt%. Then, melt spinning was carried out at 213–220 °C. Finally, hybrid fluorescent fibers were obtained, including ETPC fibers, ETPC-SA fibers, and ETPC-HQ fibers. The formation of CLCCF is explained in Figure 2. 

As shown in Figure 2, CLCC and LDPE particles were uniformly mixed, and they were subjected to shear stress and pressure stress in a mini-twin-screw extruder. All CLCC particles changed from micro-level crystals to irregular nano-level fragmentized crystals, which were well dispersed in the fiber–matrix interface (LDPE).

### 2.6. Characterizations

Transmission electron microscopy (TEM) images were obtained using a JEM-1200EX transmission electron microscope (JEOL, Tokyo, Japan). The sample was obtained by dropping the product solution onto a copper grid. The sample was placed at room temperature and atmospheric pressure to evaporate the solvent.

JSM-840 electron microscope (JEOL, Tokyo, Japan) was used to obtain SEM photographs. The obtained complex was placed in a drying oven to dry. When it was completely dried to powder, the sample was taken out for a scanning test.

X-ray photoelectron spectroscopy (XPS) spectra were obtained using a VG Scientific ESCALAB 220IXL photoelectron spectrometer. The sample was pressed and then tested. Exciting source: Al Kα ray (h*v* = 1486.6 eV), working voltage 12 kV.

FT-IR spectra were measured on a Nicolet 5700 (Nicolet Instrument Corporation, Fitchburg, MA, USA). Using the ATR method, the scanning range was 4000–500 cm^−1^. The background was collected before measurement to eliminate the interference of the substrate.

The X-ray diffraction (XRD) of the sample with the crystal structure was characterized by X-ray diffractometer (Rigaku, Tokyo, Japan). Sample preparation was carried out with an appropriate amount of dried sample, instrument voltage of 40 KV, radiation source of Cu-Kα, X-ray wavelength of 1.54056 Å, scanning range of 30–90°, and scanning rate of 0.02°/s.

UV/vis spectra were obtained on a Lambda 750S (PerkinElmer, Waltham, MA, USA) spectrophotometer at ambient temperature. A test was carried out with sample liquid and test light source for tungsten lamp and deuterium lamp. Testing wavelength range was 200–700 nm.

Fluorescent absorption and emission spectra were measured on a Cary Eclipse fluorescence spectrophotometer (Varian, San Francisco, CA, USA). The fluorescence properties of the complexes were characterized by 5 × 10 nm^2^ slit.

The fluorescence lifetime and fluorescence yield were analyzed by FLS1000 steady-state/transient fluorescence spectrometer (Edinburgh Instruments, Edinburgh, UK). Fluorescence lifetime analysis was carried out using microsecond light as excitation source on a PMT-900 detector. Fluorescence quantum yield analysis was carried out using integral sphere accessories and 450 W xenon lamp as excitation light source to obtain absolute quantum yield.

The fibers were sliced and sandwiched in copper mesh with tweezers. A high-resolution test was carried out using JEM-1200EX transmission electron microscope (JEOL, Tokyo, Japan).

The fiber strip was placed on a clean glass slide, and the solid fluorescence spectrum of the fiber was measured at room temperature. The fluorescence spectra and microscopic fluorescence of solid fibers were detected by 20/30PV Microspectral analysis integrated system (CRAIC, San Dimas, CA, USA). The excitation source was 365 nm at room temperature.

## 3. Results and Discussion

### 3.1. Adjustment of Cuboid-like Crystal Structures

The representative crystalline structures and morphological structures were investigated by TEM (Figure 1a–c) and SEM (Figure 1d–f). The samples were obtained by dropping the product solution onto a copper grid, and then they were placed at room temperature and atmospheric pressure to evaporate the solvent. It is worth noting that pure ETPC (Figure 1a–d) has an inhomogeneous size and irregular shape, showing agglomeration without the existence of stearic acid and hydroquinone. Notably, uniformly well-crystallized and mono-dispersed cuboid-like crystals can be observed (Figure 1b,c,e,f) when doped with amphiphilic molecules SA and HQ. In Figure 1b,e, ETPC-SA shows a cuboid-like crystal with an average particle size length of 2.97 μm under TEM and 3.04 μm under SEM. Similarly, ETPC-HQ has a similar crystalline structure, as shown in Figure 1c,f, appearing as a cuboid-like shape with an average particle size length of 1.52 μm under TEM and 1.59 μm under SEM. Comparing ETPC-SA and ETPC-HQ, the former is more regular and larger in shape. On the one hand, the average molecular weight of SA is two times that of HQ; on the other hand, the carboxyl groups in SA have a stronger interaction with Eu^3+^ than that of the hydroxyl groups in HQ. Hence, the above results reflect that the notion that amphiphilic molecules, such as SA and HQ, are crucial controllers of morphology. The X-ray diffraction patterns (Figure 2) show that ETPC, ETPC-SA, and ETPC-HQ have crystalline structure characteristics. SA has diffraction peaks at 2θ = 18.4° and 2θ = 20.7° (Figure 2a), and HQ has diffraction peaks at 2θ = 16.8° (Figure 2b); however, their peaks both disappear in the three complexes (as shown in Figure 2c–e), indicating that the crystal structures of SA and HQ do not exist after being doped into ETPC. It is worth noting that there is a strong and sharp diffraction peak at 2θ = 32.6°, which is the characteristic diffraction peak of ETPC, and, in another study, almost pure Phen and EuCl_3_ compounds were reacted because their peaks at 2θ = 21.4° and 2θ = 27.6° are inexistent [34]. Furthermore, the XRD patterns show that ETPC-SA and ETPC-HQ have the same crystalline characteristics as those of ETPC, indicating that SA and HQ do not interfere with the crystalline structures of ETPC, such as the crystal plane line distance and unit cells.

### 3.2. Composition Characterizations of ETPC, ETPC-SA, and ETPC-HQ

As seen in the XPS spectra (Figure 3), the peaks at 1135 eV, 531 eV, 399 eV, 285, 198 eV, and 165 eV indicate Eu3d, O1s, N1s, C1s, Cl2p, and S2p signals, respectively. As shown in Table 1, the binding energy of Eu3d in ETPC [35], ETPC-SA, and ETPC-HQ is reduced by 2.0 eV, 2.1 eV, and 2.1 eV, respectively, compared to the pure EuCl_3_ compound. The binding energy of O1s in the TTA of ETPC, ETPC-SA, and ETPC-HQ is decreased by 2.0 eV, 1.8 eV, and 1.8 eV respectively, compared to pure TTA, which is due to the increase in the polarization of the oxygen atom in the coordinated TTA.

The N1s binding energy of coordinated Phen in ETPC, ETPC-SA, and ETPC-HQ is increased by 0.8 eV, 0.9 eV, and 0.8 eV, respectively, compared to pure Phen, the reason for which is the polarization increase in the nitrogen atom in the coordinated Phen. Hence, the XPS data indicate that the oxygen atoms and nitrogen atoms coordinate with Eu^3+^ to form coordination interactions.

In Figure 4, the FT-IR spectra of SA, HQ, ETPC, ETPC-SA, and ETPC-HQ are determined within the wavenumber range of 4000–400 cm^−1^. In Figure 4a, the spectrum of SA shows its characteristic peaks at 2913cm^−1^ and 2845 cm^−1^, which correspond to the C–H stretching vibration, and at 1710 cm^−1^, which corresponds to C=O bond absorption. In Figure 4b, the IR spectrum of HQ shows bands at 3100–3300 cm^−1^, which may be assigned to the C–H stretching vibration of the benzene ring, with the peak at 1400 cm^−1^ corresponding to –OH deformation vibration and the peak at 755 cm^−1^ corresponding to the bending vibration of di-substituted benzene. In Figure 4c–e, the carbonyl stretching C=O bond of pure TTA is at 1700 cm^−1^, but it shifts to 1600 cm^−1^ in the ETPC, ETPC-SA, and ETPC-HQ complexes due to the coordination bonds formed between TTA and Eu^3+^, which results in a reduction in the strength of the C=O bond absorption. The characteristic absorption peak of C=N in pure Phen is at 1615 cm^−1^, but it shifts to 1538 cm^−1^ in the ETPC, ETPC-SA, and ETPC-HQ complexes due to the coordinated interaction between the Phen ligand and Eu^3+^. In addition, the FT-IR spectra of ETPC-SA contain a C–H signal but lose the C=O signal of SA, as shown in Figure 4d, demonstrating that SA is successfully doped and forms coordination with Eu^3+^. Similarly, in the ETPC-HQ complex, the existence of the C–H stretching vibration of the benzene ring, which has a blue-shift, and the inexistence of a –OH deformation vibration illustrate that HQ is successfully doped and forms coordination with Eu^3+^, as shown in Figure 4e. Furthermore, the lower frequency bands around 640 cm^−1^ and 580 cm^−1^ correspond to the Eu–O stretching vibration [36], and the lower frequency bands around 417 cm^−1^ are attributable to Eu–N [37]. In conclusion, the FT-IR spectra prove the existence of ETPC, ETPC-SA, and ETPC-HQ complex hybrid materials. 

### 3.3. Photophysical Properties of ETPC, ETPC-SA, and ETPC-HQ

Figure 5 displays the spectra of the UV/vis characteristic absorption peaks of SA, HQ, ETPC, ETPC-SA, and ETPC-HQ. In Figure 5a, SA has a characteristic absorption peak at 218 nm. In Figure 5b, HQ has two characteristic absorption peaks at 230 nm and 294 nm. In Figure 5c, TTA possesses a wider absorption band at around 300 nm. In Figure 5d, Phen has two peaks at 230 nm and 264 nm. The ETPC, ETPC-SA, and ETPC-HQ complexes in Figure 5e–g have three characteristic absorption peaks around 230 nm, 264 nm, and 342 nm, respectively. The peak around 230 nm belongs to the pure Phen ligand [38], and the peak at 342 nm corresponds to the singlet–singlet π–π* transitions of the TTA ligand [39]. In particular, the absorption peak of SA at 218 nm disappears in the ETPC-SA complex, which indicates that the C=O bond is coordinated with Eu^3+^. Similarly, there is no existence of the O–H absorption of HQ at 294 nm, which proves that hydroxyl causes interaction between HQ and Eu^3+^. It is interesting to note that the doping of SA or HQ has no impact on the UV characteristic absorption of the complexes.

The excitation and emission spectra of ETPC, ETPC-SA, and ETPC-HQ exhibit broad bands between 250 nm and 750 nm, as shown in Figure 6. In the excitation spectra, there are broad bands at 300–420 nm, with a distinct peak at 392 nm for ETPC, with a sharp peak at 388 nm for ETPC-SA, and with an obvious peak at 385 nm for ETPC-HQ. The emission spectra of ETPC, ETPC-SA, and ETPC-HQ show a similar peak state in the range of 500–750 nm, and they possess emission bands assigned to the ^5^D_0_ → ^7^F_J_ (J = 1–4) transitions of Eu [40], which are related to the 4f orbital electronic transitions of ^5^D_0_ → ^7^F_1_ (590.00 nm), ^5^D_0_ → ^7^F_2_, (613.80 nm), ^5^D_0_ → ^7^F_3_ (652.00 nm), and ^5^D_0_ → ^7^F_4_ (700.00 nm). The highest intensity emission peak corresponds to the ^5^D_0_ → ^7^F_2_ electron transition at 613.80 nm, which is sensitive to a changing external environment, corresponding to magnetic dipole transitions [41]. Therefore, the ^5^D_0_ → ^7^F_2_ transition will be enhanced when the external environment changes. The ^5^D_0_ → ^7^F_3_ transition is weak, as it is forbidden by the selection rules of forced dipole transitions. The ^5^D_0_ → ^7^F_1_ transition appears because of its magnetic dipole nature, while the ^5^D_0_ → ^7^F_4_ transition is weaker than the ^5^D_0_ → ^7^F_2_ transition on account of it appearing at the border of the visible spectral region [42]. It can be observed that the intensities of the excitation and emission spectra both decrease with the doping of SA or HQ compared with pure ETPC. Moreover, the intensities of the excitation and emission spectra of ETPC-SA significantly increase, which is because SA has a stronger impact on the π–π* transition in other ligands.

However, as shown in the pictures in the low right side in Figure 6a,b, when the samples of ETPC, ETPC-SA, and ETPC-HQ are exposed to a UV light resource, they emit a bright red fluorescence color. In particular, ETPC-SA emits the strongest brightness, while ETPC-HQ takes second place.

### 3.4. Morphological Structure and Dispersion of ETPC, ETPC-SA, and ETPC-HQ in Hybrid Fluorescent Fibers

The representative morphological structures of the hybrid fluorescent fibers were investigated by SEM. Figure 7a,c,e show that the diameters of the three fibers are 400 μm, 388 μm, and 390 μm. Figure 7b,d,f are enlarged from Figure 7a,c,e, respectively. ETPC fibers, in which ETPC particles are well dispersed with 3 wt% content, present a rough surface, as shown in Figure 7a,b. ETPC-SA fibers exhibit an excellent smooth surface, and ETPC-SA particles in these kinds of fibers are clearly visible with primarily mono-dispersion in fibers, as shown in Figure 7c,d. In the same situation, ETPC-HQ fibers have a comparatively smooth-looking surface compared with ETPC fibers, as shown in Figure 7e,f. Therefore, either SA or HQ can enhance the compatibility of CLCCs with the host matrix, which leads to a smoother surface on fluorescent fibers.

### 3.5. Morphological and Crystalline Structures of ETPC, ETPC-SA, and ETPC-HQ in Fluorescent Fibers 

The HR-TEM images display the interior structural details of ETPC, ETPC-SA, and ETPC-HQ in the fluorescent fibers. The samples were obtained by fluorescent fiber chips. Figure 8a,d,g demonstrate that the cuboid-like crystals in Figure 1 were smashed from 2–3 micros to several hundred nanometers through the melt-spinning process, and the shape are irregular. Figure 8b,e,h are enlarged from Figure 8a,d,g, and they display the difference in interface between the complex crystal and matrix LDPE in the fibers. In the images, it seems that the dark black area contains CLCCs. We can find a blurry interface in Figure 8e, which indicates a stronger interaction between the ETPC-SA crystal and the host matrix fiber, as well as better dispersion of the fluorescent crystal. Figure 8c,f,i reveal that smashed crystals still present a well-ordered atomic arrangement, and the distances between the adjacent crystal planes are about 0.308 nm for ETPC, 0.254 nm for ETPC-SA, and 0.320 nm for ETPC-HQ. These results indicate the following: (1) SA and HQ do not interfere with the ordered atom arrangement, but they help to adjust the crystal shape from irregular to regular as shown in Figure 1; (2) SA possesses a better function for both crystal shape (as shown in Figure 1 and Figure 8) and dispersion in the LDPE host (as shown in Figure 8). Thus, we think that ae flexible C–C link structure, such as SA, provides better conformation changeability [43].

### 3.6. Luminescent Property of ETPC, ETPC-SA, and ETPC-HQ in Hybrid Fluorescent Fibers

In Figure 9, the emission spectra and photofluorogram of the solid-state fluorescent fibers, in which the mass ratio of the complex occupying the host matrix (LDPE) is about 3 wt%, are tested by the 20/30PV Microspectral analysis integrated system (CRAIC, San Dimas, CA, USA) and excited by 365 nm. Meanwhile, the contrast of the fluorescent fibers excited by ultraviolet light with fiber under sunlight is also exhibited. Compared with the emission spectra of the CLCCs (as shown in Figure 6), CLCC fibers similarly possess ^5^D_0_ → ^7^F_J_ (J = 0–4) transitions [44] and an excellent luminous property. The five 4f orbital electronic transitions are ^5^D_0_ → ^7^F_0_ (580.00 nm), ^5^D_0_ → ^7^F_1_ (590.00 nm), the hypersensitive ^5^D_0_ → ^7^F_2_ (613.00 nm), ^5^D_0_ → ^7^F_3_ (650.00 nm), and ^5^D_0_ → ^7^F_4_ (700.00 nm). We can say that the host matrix can supply a remarkable carrier environment, which not only holds the luminescence property but also protects the CLCCs from environmental erosion and damage.

In Figure 9a–c, the photofluorogram of the fluorescent fibers was obtained using a 15× magnification factor from a fluorescence microscope. In the fluorescent fibers, CLCC particles could easily be discovered, and they were dispersed well in the fibers. 

Figure 9d–i demonstrate the contrast of fluorescent fibers under ultraviolet light and under sunlight. Both visible fibers possess smooth and uniform morphology structures, and they emit a strong red light, which provides a basis for wider applications of fluorescent fibers. 

### 3.7. Fluorescence Lifetime and Fluorescence Quantum Yield of CLCCs and CLCCF

The fluorescence lifetime decay curve and the fluorescence quantum yield of CLCCs and CLCCF were obtained at an excitation wavelength of 365 nm and emission wavelength of 613 nm. The results of each sample were tested three times, and each time, the results of each test were the same, as shown in Figure 10 and Table 2, which present the significant parameters characterizing the photoluminescent property. The ETPC-SA fiber and ETPC-HQ fiber have three fluorescence lifetimes, and the curves are very well fitted to a triple-exponential function. The others have two fluorescence lifetimes, and the curves are very well fitted to a double-exponential function. The average fluorescence lifetimes (τ_A_) of all luminous materials in this report, as shown in Table 1, were calculated according to the following formula: τ_A_ = (τ1 × B1% + τ2 × B2% + τ3 × B3%)/100%. Not only does ETPC-SA, adjusted by stearic acid, generate the best photo emission property with a quantum yield of 98.4%, but it also owns the longest average luminescent lifetime, 785µs, amongst the complexes in our work. 

The average fluorescence lifetimes of ETPC fibers, ETPC-SA fibers, and ETPC-HQ fibers are 1259 µs, 840 µs, and 1189 µs, respectively, which are higher than those of the complexes. However, CLCCF has a notably lower quantum yield than the complexes, illustrating that the fiber–matrix interface (LDPE) can distinctly prolong fluorescence lifetime but weaken fluorescence quantum yield, especially for ETPC-HQ. Even so, we can say that the excited electrons standing in the excited state for a relatively long time would increase the afterglow property of the fluorescent fibers.

### 3.8. Bright Red Fabric Applications

Flexible fluorescence emission materials, emission fibers, and related wearable fabrics with a bright emission property are increasingly becoming excellent candidates for wearable devices with the development of modern society.

Figure 11 exhibits ropes, blankets, fabrics, and novel hairpiece applications, which were made from CLCCF. The colors of these fabrics vary depending on whether they are under sunlight or ultraviolet excitation. With the CLCC powder content in CLCCF increasing from 0.5 wt% to 5 wt%, the colors of the final CLCCF gradually deepen and become redder. Furthermore, CLCCF in this report has the same wear performance as that of ordinary fibers. It is its luminescent composition that differs from the radioactive zinc sulfide luminous substance, and there is no existence of phosphorus, lead, chromium, potassium, or other hazardous heavy metals. It is also different from a variety of reflective materials, so we do not need other materials to coat textile superficies.

For applications, they can be made as luminous clothing; luminous rope; thread and belts; nets; carpets; and household articles in some fields, such as building decoration, transportation, aviation and navigation, night operations, fire emergency services, daily life warnings, and entertainment clothing. It is interesting that a polymer hairpiece can have a high vivid degree and color variability as a result of doping different CLCCs, and such a luminous hairpiece can be worn as a headdress by bald or hairless people, as part of a costume, as official or professional attire, or as part of a fashionable decoration. Moreover, luminous hairpieces can help people discern whether they are in a high-ultraviolet environment.

CLCCF in this report emits red luminosity well and possesses remarkable advantages, including energy saving, environmental protection, safety, being maintenance free, and offering a wide application range. We can declare that CLCCF is an excellent “safe red light source”.

## 4. Conclusions

In summary, the results mentioned in this article regard the obtained cuboid-like ETPC-SA and ETPC-HQ crystalline complexes for which the shape of crystals can be adjusted by amphiphilic molecules, such as SA and HQ. The TEM images revealed that novel ETPC-SA and ETPC-HQ had more regular morphology after amphiphilic molecules were incorporated into ETPC crystalline. To obtain CLCCF, the ETPC-SA and ETPC-HQ crystals were doped into a host matrix (LDPE) using a twin extruder and then melt spinning. In the extruding process, the regular cuboid-like crystals were deformed and smashed but the crystalline structures still presented well-ordered atomic arrangement. This result shows that SA and HQ do not interfere with the ordered atom arrangement but, rather, help to adjust the crystal shape from irregular to regular. SA behaves well for both crystal shape and dispersion in the LDPE host, which leads to a notable improvement in the compatibility between CLCCs and the fiber–matrix interface. However, ETPC-SA has the best photon emission property with a quantum yield of 98.4%, as well as the longest average excited electron life of 785 µs within complexes. However, fibers carrying HQ, i.e., ETPC-HQ fibers, have an average excited electron life of 1189 µs, but they also have the lowest quantum yield, which means that the fiber–matrix interface incorporated with HQ could increase the afterglow property of fluorescent fibers. Finally, bright fluorescent red fabrics have flexibility and smooth hand-feeling properties. We expect these fabrics to be used in building decoration, aviation, navigation, night operations, fire emergency services, daily life, entertainment clothing fields, and so on.

## Data Availability

All data, models and codes generated or used during the study appear in the submitted article.

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
