# Peer review of "Novel Cuboid-like Crystalline Complexes (CLCCs), Photon Emission, Fluorescent Fibers, and Bright Red Fabrics of Eu3+ Complexes Adjusted by Amphiphilic Molecules"

_polymers, 2022, doi:10.3390/polym14050905_

Round 1

Reviewer 1 Report

The manuscript characterizes by high level of scientific soundness and general significance for materials chemistry. I think that it deserves publication in Polymers (MDPI).

My only criticism is the level of characterization of the complex Eu-TTA-Phen.

Do the authors consider this so-called complex as individual compound or just as a mixture of the components suitable for further incorporation to the polymer matrix?

If the author reckon that they deal with individual compound, they are obliged to investigate at least elemental ratio (C : H : N : Eu) in order to confirm proposed general structure depicted on the Scheme 1.

Reviewer 2 Report

General Comment:

The manuscript from Q. Tang, S. Liu, J. Liu, Y. Wang, Y. Wang, S. Wang, Z. Du, L. Huang, L. A. Belfiore, J. Tang presents the study of novel cuboid‑like crystalline complexes. So the authors present three new complexes: Eu3+-TTA-Phen (ETPC), Eu3+-TTA-Phen-SA (ETPC-SA) and Eu3+-TTA-Phen-HQ (ETPC-HQ). The authors have realized the synthesis and the characterizations by powder XRD diffraction, XPS, TEM, SEM, IR and UV absorption and photoluminescence spectroscopies from their compounds. In a second step, hybrid fluorescent fibers containing these different compounds were also manufactured and studied. All characterizations are well conducted and are in adequacy with the journal “Polymers”. Some improvement could be realized before that this manuscript is published. I recommend publication after minor revisions.

Comments and Minor points:

1- Characterizations part: More specifically for the quantum yield, more detail should be indicated (method, absolute or relative measurement,…). For the lifetime measurements, no information are given (excitation, setup, source,…). More detail should be also indicated for powder diffraction (lambda,…), IR (ATR or not,…), XPS...

2- No crystal structure is presented (No single crystal suitable for XRD was obtained?) and the powder diagrams are weak with very few diffraction peaks, how can the authors be sure that their compounds are pure? Elemental analyses or an EDS analyzes would help to confirm the purity of the phases.

3- It seems that there are two small errors on the energy calculations: for O1s the energies are 1.8 and 1.8 instead of 2.2 and 2.2 eV.

4- It could be useful for the understanding to specify if the spectroscopic measurements or others characterizations are made on solid, liquid or suspension. For excitation and emission spectra (Figure 6 and 9), it is generally more common to specify the emission wavelength for excitation spectra and the excitation wavelength for emission spectra. For figure 9, lambda emission is not necessary, the spectra were recorded under 365 nm excitation. For Eu(III), the transition at 530 nm attributed to 5D0-7F0 is very surprising, are the authors sure of their attrition for this band? Usually and as observed on hybrid fibers this transition is observed around 580-590 nm.

5- The lifetime measurements have it been measured several times? The values must be indicated with uncertainty errors. For the quantum yields, how they are measured? Which is the reproducibility? What is the uncertainty? In the Table 2, the seventh line is redundant, it could be deleted.

6- What is the thermal stability of these compounds? The analysis of the thermal stability could be an important information on the field of application of the compounds and their use as luminescent materials.

Author Response

请参考附件

Reviewer 3 Report

 The manuscript presents an experimental work on the production and characterization of flexible fluorescent fibers for wearable fabrics based on cuboid-like crystals of Eurobium complexes which can be processed by common fabricating methods. The work is well written and presented but needs some revisions to improve its scientific impact.

My comments are listed below

  1. In the introduction the meaning of “vicinity environmental design for organic molecules” is unclear and needs to be explained.
  2. The authors should explain why cuboid-like crystals can better disperse than spherical crystals in a polymer matrix.
  3. In the Experimental section, only the instruments used are mentioned. The authors should provide more details on the measurement conditions used for each characterization.
  4. Figure 1 reports only one measurement on one cuboid-like crystal. The authors should provide the average values and standard deviation on a significant number of crystals measured in different images.
  5. Figure 1: please change the scale bar “um” with “µm” using the Greek letter.
  6. Figure 7: please provide image analysis and size distribution histogram for the fiber diameter. The authors should report the number of fibers they used for measuring the average diameter of the fibers.
  7. The authors should explain how they measured the lifetime fluorescence and how they calculate the number of lifetime fluorescence. It is not clear in the manuscript how some materials had 3 luminescence points and the other two
  8. The authors should add some consideration on the change of thermal or mechanical properties of polymer matrix after the addition of the Eurobium complexes

Round 2

Reviewer 3 Report

The authors addressed the comments. The paper is now suitable for publication.